# Development of a Sericin Hydrogel to Deliver Anthocyanins from Purple Waxy Corn Cob (*Zea mays* L.) Extract and In Vitro Evaluation of Anti-Inflammatory Effects

**DOI:** 10.3390/pharmaceutics14030577

**Published:** 2022-03-06

**Authors:** Nattawadee Kanpipit, Natsajee Nualkaew, Worawikunya Kiatponglarp, Aroonsri Priprem, Suthasinee Thapphasaraphong

**Affiliations:** 1Biomedical Science Program, Graduate School, Khon Kaen University, Khon Kaen 40002, Thailand; natawadee.k@kkumail.com; 2Department of Pharmacognosy and Toxicology, Faculty of Pharmaceutical Sciences, Khon Kaen University, Khon Kaen 40002, Thailand; nnatsa@kku.ac.th; 3Synchrotron Light Research Institute (Public Organization), Nakhon Ratchasima 30000, Thailand; worawikunya@slri.or.th; 4Faculty of Pharmacy, Mahasarakham University, Maha Sarakham 44150, Thailand; aroonsri@kku.ac.th; 5Department of Pharmaceutical Chemistry, Faculty of Pharmaceutical Sciences, Khon Kaen University, Khon Kaen 40002, Thailand

**Keywords:** sericin-alginate hydrogels, FT-IR, controlled release, inflammatory cytokines, topical hydrogels

## Abstract

Sericin-alginate hydrogel formulations with purple waxy corn (*Zea mays* L.) cob extract (PWCC) for topical anti-inflammatory application are developed and evaluated. The physical properties such as viscosity, pH, and anthocyanin release are examined and in vitro anti-inflammatory activities, such as NO inhibition and IL-6, IL-1β, TNF-α, iNOS, and COX-2 expression, are evaluated in LPS-stimulated RAW 264.7 murine macrophages. The sericin-alginate hydrogel is prepared by physical crosslinking through the ionic interaction of the polymers combined with anthocyanin from PWCC at pH 6.5. The hydrogel formulation with 2.00% *w*/*v* sericin, 0.20% *w*/*v* alginate, and 0.15% *w*/*v* PWCC (SN6) shows a suitable viscosity for topical treatment, the highest nitric oxide inhibition (79.43%), no cytotoxicity, and reduced expression of IL-6, IL-1β, and TNF-α mediators. Moreover, the SN6 formulation displays a sustained anthocyanin release over 8–12 h, which correlates with the Korsmeyer–Peppas model. The FT-IR spectrum of SN6 confirmed interaction of the sericin polymer with anthocyanins from PWCC via H-bonding by the shifted peak of amide I and amide III. In addition, the anthocyanin is stable in sericin hydrogels under heating-cooling storage conditions. Therefore, we suggest that this hydrogel formulation has potential as an anti-inflammatory agent. The formulation will be further investigated for in vivo studies and clinical trials in the future.

## 1. Introduction

Skin is the first barrier for protection from the external environment. Dermal macrophages release inflammatory mediators including tumor necrosis factor-α (TNF-α) and interleukin (IL)-6 and IL-1β that are involved the inflammatory immune response and wound healing [1]. Plants contain active agents such as flavonoids, phenolics, triterpenes, cinnamic acid, and anthocyanins that have anti-inflammatory properties [2]. Therefore, it is interesting to develop natural plant extracts as topical formulations for anti-inflammatory products.

Hydrogels have been widely used for biomedical applications such as controlled drug release, biocompatibility, and increasing drug stability, as well as providing a moist environment and helping to reduce inflammation during skin wound healing [3]. Hydrogels are hydrophilic polymer chains with a 3D-crosslinked structure, a high-water content, and good biodegradability, biocompatibility, and drug delivery properties [4]. In addition, natural and synthetic polymers have been combined to produce a hydrogel network called biosynthetic polymeric material, which is considered an emerging class of polymeric networks. This material was demonstrated to improve mechanical and thermal properties and biocompatibility compared to those synthesized from single components [5].

Sericin is a natural protein waste from *Bombyx mori* cocoons generated during the silk production process. Sericin constitutes about 25% of the total silk protein and contains polar amino acids such as serine (30–33%), aspartic acid (17–20%), and glycine (16%) [6]. Sericin has been reported for its anti-inflammatory properties, suppressing the generation of inflammation mediators such as TNF-α and interleukin-1β (IL-1β) [7]. Sericin has been extensively used as a biomaterial as it has good biocompatibility and biodegradability. It has been used to produce biological scaffolds through co-polymerization or simple crosslinking with other polymers to improve their physical and drug release properties [8].

Alginate is a natural polymer from brown algae composed of blocks of mannuronic and guluronic acids with good biocompatibility and biodegradability. Alginate has been applied as a bioadhesive and to increase viscosity in pharmaceutical products [9]. Sericin and alginate have been previously used for a diclofenac sodium delivery system with improved sustained release properties [10]. Therefore, sericin and alginate are good candidates for combination in a hydrogel with improved active compound loading and controlled release properties.

Polyvinyl alcohol (PVA) is a simple, linear, hydrophilic synthetic polymer with biocompatibility and biodegradability properties suitable for controlled-release drug delivery in pharmaceutical applications [5]. It has been used to form interpenetrating polymer network (IPN) hydrogels with other synthetic or natural polymers such as alginate or sericin, to successfully balance the hydrogels’ qualities [5,11].

Anthocyanins are natural pigments found as red, pink, purple, and blue colors in plants. Malvidin-3-glucoside and malvidin-3-galactoside obtained from blueberry fruits caused anti-inflammatory effects [12]. Purple waxy corn (*Zea mays* L.) is a natural enriched source of anthocyanins [13]. Cyanidin-3-glucoside (C3G), pelargonidin-3-glucoside (Pg3G), and peonidin-3-glucoside (Pn3G) were the major anthocyanins present in purple waxy corn extracts, of which C3G had the highest content [14]. Previous studies have shown that C3G has anti-inflammatory [15] and antioxidant activities [16] that are suitable for biomedical product applications. However, temperature and pH can affect the stability of anthocyanins. Therefore, it is essential to develop products that prevent anthocyanin degradation to improve the therapeutic potential of anthocyanins at target sites [17].

In this study, we have developed a hydrogel formulation to deliver anthocyanins from natural sources. Purple waxy corn (*Zea mays* L.) cob extract, a natural enriched source of anthocyanins, was utilized as the model for the hydrogel formulations.

Hydrogel formulations combining sericin as a natural polymer with polyvinyl alcohol (PVA) and sodium alginate and containing purple waxy corn cob extract were developed and evaluated for their stability and active ingredient releasing properties. This product was developed for potential topical anti-inflammatory applications. The physical and biological properties were characterized related to anti-inflammatory activity.

## 2. Materials and Methods

### 2.1. Materials

Dimethyl sulfoxide (DMSO) and 3-[4,5-dimethylthiazol-2-yl]-2,5-diphenyltetrazolium bromide (MTT) were obtained from Thermo Scientific, (Waltham, MA, USA). Lipopolysaccharide (LPS), Nomega-Nitro-L-arginine methyl ester hydrochloride (L-NAME), *N*-(1-naphthyl)ethylenediamine, sulfanilamide, lipopolysaccharide (LPS), and phosphoric acid were obtained from Sigma-Aldrich (St. Louis, MO, USA). Polyvinyl alcohol (PVA) was obtained from Chem-Supply Pty Ltd. (Gillman, Australia). Sodium carbonate (Na_2_CO_3_), sodium hydrogen carbonate (NaHCO_3_), disodium hydrogen phosphate (Na_2_HPO_4_), potassium dihydrogen phosphate (KH_2_HPO_4_), and sodium chloride (NaCl) were obtained from Ajax Finechem Pty Limited (Australia). Dulbecco’s modified Eagle’s medium (DMEM), fetal bovine serum (FBS), penicillin-streptomycin (10,000 U/mL), and phosphate-buffered saline (PBS) were obtained from Gibco (Gaithersburg, MD, USA).

### 2.2. Preparation of Extracts

#### 2.2.1. Extraction of Sericin from Silkworm Cocoons

The silkworm cocoons (J108 strain) were obtained from Queen Sirikit Sericulture Center (Khon Kaen, Thailand) in January 2020. Cocoons were aged 42 days at collection. The silkworm cocoon was cut into small square pieces of 1 × 1 cm. Twenty-four grams of cocoon materials were added to 700 mL of purified water and autoclaved (Tomy-SX-700, Tokyo, Japan) at 120 °C, 2.5 bar for 60 min. After filtration through Whatman No. 1, the filtrate was freeze-dried at −85 °C, 0.595 bar (Labconco, Kansas, MO, USA) to obtain the sericin extract [18]. The sericin extract was stored at 4 °C before use. The protein content was determined by Bradford assay and molecular weight profiles were determined by polyacrylamide gel electrophoresis (SDS-PAGE) [18].

#### 2.2.2. Extraction of Anthocyanin from Purple Waxy Corn Cobs

The purple waxy corn cobs (Thai purple waxy corn) were obtained from the Plant Breeding Research Center for Sustainable Agriculture (Khon Kaen University, Thailand). The cobs were ground twice, with a hammer mill and then a blender, to obtain cob powder. The cob powder was macerated in 50% ethanol (ratio of powder to solvent 1:25) with stirring at 25 °C for 24 h. The filtrate was collected. The residue was then reextracted twice following the same procedure. The filtrate was evaporated and then freeze-dried at −85 °C, 0.595 bar (Labconco) to obtain the crude extract (purple waxy corn cob extract, PWCC). PWCC was stored protected from the light at 4 °C. The anthocyanin content was determined by pH-differential method, in which the total monomeric anthocyanin content was determined based on the structural change of the anthocyanin chromophore between pH 1.0 and 4.5 [19].

### 2.3. Preparation of Sericin-Alginate Hydrogels Containing Anthocyanin from Purple Waxy Corn Cob Extract

Twelve formulations of sericin-alginate hydrogels (SN1–SN12) were prepared from the stock solutions of 5% *w*/*v* sericin solution, 10% *w*/*v* PVA, 5% *w*/*v* alginate in phosphate buffer pH 6.5 and 10% *w*/*v* PWCC in 50% EtOH in phosphate buffer pH 6.5. The stock solutions were mixed to give final concentrations of 2.00% *w*/*v* sericin, 4.00% *w*/*v* PVA, 0.00, 0.20, 0.30, and 0.40% *w*/*v* alginate, and 0.00, 0.15, and 0.50% *w*/*v* PWCC according to Table 1. The final volume was adjusted to 5 mL with phosphate buffer pH 6.5 and the mixture was stirred for 30 min at room temperature for each hydrogel formulation.

### 2.4. Characterization of Sericin-Alginate Hydrogels

All formulations were evaluated for their characterization such as appearance by visual inspection, pH of the formulations by pH meter, and total anthocyanin and viscosity were determined using a Brookfield viscometer (Model DV–III, Middleboro, MA, USA) at 100 rpm with spindle no. 64 in triplicate. The total anthocyanin content of all formulations was also determined by the pH-differential method [19]. 

### 2.5. In Vitro Release of Anthocyanins

The in vitro release of anthocyanins from the formulations was determined using a Franz diffusion cell with a cellulose acetate membrane filter (pore size 0.45 μm, diameter 25 mm, Filtrex, Encinitas, CA, USA) at 37 °C. Phosphate buffer pH 5. 5 was the sink condition in the receptor chamber. One-gram aliquots of the formulations were added to the donor chamber. Samples were withdrawn from the receptor cell after 0.5, 1, 2, 4, 6, 8, and 12 h and total anthocyanin content was determined by the pH-differential method [20].

### 2.6. Release Kinetics Study

The release kinetics of the formulations were studied using the following four kinetic models: (1) cumulative percentage of release and time (zero-order), (2) log of cumulative percentage of release and time (first-order), (3) cumulative percentage of release and square root of time (Higuchi model), and (4) log of cumulative percentage of release and log of time (Korsmeyer–Peppas semi-empirical model). Data was analyzed by linear regression using the best linear correlation coefficient (r) [20].

### 2.7. Scanning Electron Microscopy (SEM) Analyses

Hydrogels were dried by vacuum drying ovens (Loboao, LDZ24T, Shanghai, China). The cross-sectional morphologies of the dry hydrogels were examined using scanning electron microscopy (FEI Quanta 450, Sunrise Valley Drive Reston, VA, USA) at an operating voltage of 10 keV. Segments of cross-sectional surfaces were prepared by the slicing method and then sprayed with gold for SEM measurements. The pore sizes were determined by measuring 50 random pores from SEM images of the same sample using ImageJ software /Fiji software (National Institutes of Health NIH, MD, USA).

### 2.8. Swelling Studies

The dried hydrogel films were cut into the size 1 × 1 cm and accurately weighted (M_d_). The hydrogel films were then immersed in PBS pH 7.4 in the culture plate at room temperature for 48 h. The swollen films were removed from plate, excess buffer was wiped off with the filter paper, The weights of the infiltrated hydrogels (M_w_) were measured at regular intervals (at 0.5, 1, 2, 4, 6, 8, 12, 24, and 48 h). The percentage swelling was calculated as the following equation [21]:%Swelling = [(M_w_ − M_d_)/M_d_] × 100

M_w_ = The weight of infiltrated hydrogelsM_d_ = The weight of dried hydrogel

### 2.9. Anti-Inflammatory Study

#### 2.9.1. Cytotoxicity Assay

The cytotoxicity of the formulations was determined in the RAW 264.7 murine macrophage cell line (American Type Culture Collection, Manassas, VA, USA) cultured in DMEM supplemented with 10% FBS, 100 µg/L streptomycin, and 100 IU/mL penicillin at 37 °C and 5% CO_2_. Cells were seeded into 96-well plate at an initial concentration of 1 × 10^4^ cells/well in culture medium then incubated for 24 h at 37 °C and 5% CO_2_. The media was aspirated and replaced by test samples. The test samples were hydrogel formulations (SN1–SN12), 0.15% *w*/*v* PWCC (PWCCS0.15), 0.50% *w*/*v* PWCC (PWCCS0.5) or 2.00% *w*/*v* sericin in culture media (1:10 ratio) to give a total volume of 100 μL. The media from non-treated cells was a negative control ((−) control). After 24 h incubation at 37 °C and 5% CO_2_. Cells were then incubated for 24 h at 37 °C and 5% CO_2_ After the removal of media from the 96-well plates, cell viability was assessed by MTT (3-(4, 5-dimethyl thiazol-2yl)-2, 5-diphenyl tetrazolium bromide) reduction assay with formazan absorbance measured at 570 nm using a microplate reader [22]. The cell viability was calculated as follows: % cell viability = (absorbance_sample_/absorbance_(__−)control_) × 100.

#### 2.9.2. Inhibition of Nitric Oxide (NO) Production

RAW 264.7 cells were seeded into 96-well plates at an initial concentration of 1 × 10^4^ cells/well and incubated for 24 h at 37 °C with 5% CO_2_. Then, the media was aspirated and replaced by test samples. The test samples were hydrogel formulations (SN1–SN12), 0.15% *w*/*v* PWCC (PWCCS0.15), 0.50% *w*/*v* PWCC (PWCCS0.5), 2.00% *w*/*v* sericin or 200 µM L-NAME (as a positive control) in culture media (1:10 ratio) to give a total volume of 100 μL. After 1 h incubation at 37 °C and 5% CO_2_, cells were stimulated with of 1 µg/mL lipopolysaccharide (LPS) for 24 h at 37 °C and 5% CO_2_. Then 100 µL of the culture media was collected and mixed with 100 µL of Griess reagent (0.1% N-(1-Naphthyl) ethylenediamine and 1% sulfanilamide in 5% phosphoric acid). The absorbance at 540 nm was measured using a microplate reader [22]. The media from non-treated cells was used as a negative control ((−)control). Cell viability was assessed by MTT assay. The nitric oxide inhibition was calculated as a percentage, %NO inhibition = (absorbance_control_ − absorbance_sample_)/absorbance_control_) × 100.

### 2.10. Real-Time Quantitative Reverse Transcription-PCR

Total RNA was isolated from RAW 264.7 cells using TRIzol reagent (Invitrogen Life Technologies, Carlsbad, CA, USA) according to the manufacturer’s instructions. Two hundred nanograms of total RNA were used to synthesize complementary DNA (cDNA) using M-MLV reverse transcriptase (Promega, Madison, WI, USA) following the manufacturer’s instructions. Gene expression levels were determined by qRT-PCR using an Applied Biosystems 7500 fast real-time PCR system (Applied Biosystems, Bedford, MA, USA). The reaction mixture was prepared using PowerUp^TM^ SYBR Green Master Mix (2×), 10 μM forward and reverse primers (TNF-α, IL-1, IL-6, iNOS, COX-2), and cDNA template (50 ng) and adjusted to a final volume of 10 μL with nuclease-free water. The qRT-PCR conditions were initial UDG activation for 1 cycle at 50 °C for 2 min, Dual-Lock^TM^ DNA polymerase for 1 cycle at 95 °C for 2 min, PCR amplification for 40 cycles with denaturation for 1 cycle at 95 °C for 15 sec, and annealing and extension at 60 °C for 1 min. The GAPDH gene was used as an endogenous control reference gene. The specific PCR amplifications were assessed by melting curve and the level of gene expression was determined by the ΔΔ^−CT^ method using 7500 software v2.3 (Applied Biosystems, Bedford, MA, USA) [23].

### 2.11. Measurement of Inflammatory Cytokine Levels

Inflammatory cytokine levels were measured in RAW 264.7 cells. TNF-α and IL-6 levels were determined in cell culture media and IL-1β cytokine level was determined in cell lysate using enzyme-linked immunosorbent assay (ELISA) according to the manufacturer’s protocols (Abcam, Biomed Diagnostics, Cambridge, MA, USA).

### 2.12. Fourier Transform Infrared Spectroscopy FT-IR Spectroscopy

Hydrogel films were transferred to a petri dish and dried in a vacuum oven for 24 h [21]. FT-IR determination was performed by Bruker Tensor 27 FT-IR spectrometer with an ATR cell for the spectral region of 400–4000 cm^−1^ [20] and compared to FT-IR spectra of PWCC extract, sericin and alginate.

### 2.13. Stability Evaluation

The stability of the formulations was determined by thermal stability test using heating/cooling cycle. The formulations were placed alternately in a fridge at 4 °C and an oven at 45 °C every 24 h, for 7 days [24]. The hydrogel formulations were evaluated for total anthocyanin content, viscosity, and pH before and after storage.

### 2.14. Data Analysis

Data were analyzed using SPSS version 28 (SPSS Inc., Chicago, IL, USA; licensed KKU software). Data are presented as mean ± standard error of mean (SEM). Differences between groups were assessed by one-way analysis of variance (ANOVA) followed by Dunnett’s test for post hoc testing. The comparison in stability test was evaluated by paired sample *t*-test. A *p*-value of less than 0.05 was considered statistically significant.

## 3. Results and Discussion

### 3.1. Extraction of Sericin by High Temperature and High Pressure (Autoclave) Technique

This extraction method by high temperature and high pressure (autoclave) technique is simple and can be performed without using organic solvents in the process, which is suitable for industrial applications. From a previous study, this method could provide a yield of 8.41–11.60% *w*/*v* [25]. Our sericin extract yield was 10.50% *w*/*w*. The total protein content determined by a Bradford assay was 29.28 ± 0.50 mg per g dry extract. The molecular weight of sericin in our extract was estimated by comparison with a molecular weight marker (Precision Plus Protein^TM^ Blue Unstained Protein Standard, Strep-tagged recombinant in the MW range of 10–250 KDa) Bio-Rad, CA, USA) using SDS-PAGE The result showed the molecular weight of sericin ranged from 37 to 250 kDa (Appendix A). While sericin extract by high temperature and high pressure (autoclave) technique by Aramwit et.al., (2010) indicated MW in the wide range of 20–220 kDa [18]. The sericin obtained from different extraction methods provides different MW, properties, and activities [6], which affect the hydrogel formation properties. However, the high MW polymer might be suitable to perform hydrogel network. The high pressure and high temperature (autoclave) extraction method is the simple way, solvent-free technique with no reagent consumption, which might be suitable to obtain the high MW and wide range of sericin. However, sericin extracted by autoclave might provide MW range variation depending on the condition. Therefore, temperature, pressure, extraction time, and variety of silkworm cocoons should be considered. To control the quality of sericin raw materials, characterization of the crude sericin extract such as physical and chemical properties (MW range determination, protein content, spectroscopy evaluation, etc.) is necessary before hydrogel preparations [25].

### 3.2. Extraction of Anthocyanin from Purple Waxy Corn (Zea mays L.)

The extraction of anthocyanin from purple waxy corn cob by 50% *v*/*v* ethanol in water showed an extraction yield of 10.01% *w*/*w*. The total anthocyanin content was obtained at 3.79 ± 0.51 mg of cyanidin-3-glucoside equivalents (C3GE) per gram of dry weight. A previous study reported the highest anthocyanin content was 2.42 ± 0.03 mg of C3GE per gram of dry weight from purple waxy corn cobs extracted with 50% ethanol [14].

### 3.3. Characterization of Sericin-Alginate Hydrogels

The purple waxy corn cob hydrogels were found to have a homogenous texture with good consistency and a pleasant appearance by visual inspection. The color of formulations ranged from no color to light or deep purple depending on the amount of anthocyanin in the hydrogel because the flavylium cation in anthocyanins becomes deprotonated to form a violet colored quinonoidal base in neutral conditions (Figure 1) [26]. The pH of all formulations was between 6.45 and 6.71 (Table 2), which is compatible with normal human skin with no irritation [27]. However, the SN9 to SN12 hydrogels containing 0.50% *w*/*v* PWCC showed a deep purple color, which is considered to be not visually appealing in skin care products (Figure 1).

### 3.4. Determination of Total Anthocyanin Content in Sericin-Alginate Hydrogel Formulations

The total anthocyanin content of the sericin hydrogel formulations was determined by the pH-differential method. The 0.15% *w*/*v* and 0.50% *w*/*v* PWCC extracts were found to contain a total anthocyanin content of 43.75 ± 1.44 and 101.93 ± 19.54 mg C3GE/L, respectively. Comparison of the anthocyanin content in each PWCC hydrogel with the corresponding purple waxy corn cob extract solutions (PWCCS) at 0.15% *w*/*v* or 0.50% *w*/*v* showed no significant difference for the same concentration (Table 2). The hydrogel formulations were adjusted to pH 6.5 to stabilize the anthocyanins [26].

### 3.5. Viscosity of Hydrogel Formulations

The viscosities of the hydrogel formulations containing 0.15% PWCC (SN5–SN8, range 35.39–37.55 Pa·s) and 0.50% PWCC (SN9–SN12, range 34.27–36.52 Pa·s) were lower than their equivalent base hydrogel formulations without PWCC (SN1–SN4, range 35.52–39.55 Pa·s), as shown in Table 2. Increasing concentrations of alginate (0.20–0.40% *w*/*v*) also increased the viscosity in a dose-dependent manner (Table 2) due to the gelling property of alginate. Therefore, SN4, SN8, and SN12 provided the highest viscosity for each group at 39.55, 37.55, and 36.52 Pa·s, respectively. In addition, increasing the concentration of PWCC from 0.15% to 0.50% *w*/*v* caused a reduction in viscosity.

Viscosity is an important property of skin products and should be optimized to improve controlled release properties, stability of the formulation, and easy application on the skin surface [27]. However, to select the suitable formulation, all formulations were subjected to further studies of their release profiles and the stability of PWCC. The suitable formulation will be selected from the formulations that provided good control releasing and a high stability of PWCC. The viscosity of formulations could affect anthocyanin release and the swelling properties of the hydrogel. Formulations with high viscosity have been found to have lower diffusion rates and slower release than formulations with low viscosity due to the presence of increased cross-linked networks in hydrogels with high viscosity [28].

### 3.6. Anti-Inflammatory Properties of Hydrogel Formulations

#### 3.6.1. Cytotoxicity on RAW 264.7 Cells

All hydrogel formulations (SN1–SN12), 2.00% sericin, PWCCS0.15, and PWCCS0.5 caused no cytotoxicity on both untreated RAW 264.7 cells (Appendix A) and LPS-treated RAW 264.7 cells (Figure 2).

Moreover, some formulations, such as SN2, SN3, SN5, SN6, and SN9, could enhance LPS-treated RAW 264.7 cell proliferation more than sericin. Especially, the SN6 formulation with 0.15% PWCC provoked significantly higher cell proliferation on LPS-treated RAW 264.7 cells than other formulations, including the equivalent formulation without PWCC (SN2), as shown in Figure 2.

#### 3.6.2. Inhibition of Nitric Oxide (NO) Production by LPS-Stimulated RAW 264.7 Cells

The cell viability of LPS-stimulated RAW 264.7 cells treated with sericin, PWCCS, and alginate are indicated in Appendix A. Non-LPS-treated RAW 264.7 cells (negative control) showed 100% cell viability and LPS-stimulated RAW 264.7 cells showed reduced cell viability of 86.42%. PWCC and alginate (0.01–0.50% *w*/*v*) and sericin (0.01–2.00% *w*/*v*) caused no cytotoxicity. In addition, the dose-dependent effects of sericin, PWCCS, and alginate on nitric oxide inhibition are presented in Appendix A.

The hydrogel formulations with PWCC extract (SN5–SN12) exhibited higher nitric oxide inhibition than formulations without PWCC extract (SN1–SN4) in LPS-stimulated RAW 264.7 cells (Figure 3). Formulation SN6 exhibited the highest nitric oxide inhibition of the formulations, with the percentage of nitric oxide inhibition as high as the positive control (L-NAME). Therefore, SN6 might have the ability to enhance nitric oxide inhibition and anti-inflammatory functions.

LPS stimulation of RAW 264.7 cells induced nitric oxide synthase (iNOS), an enzyme responsible for promoting the inflammation process via NO stimulation. Therefore, iNOS is one of the target mechanisms of anti-inflammatory agents [29]. A previous study found the anti-inflammatory activity of purple rice cultivar extracts containing the anthocyanin cyanidin 3-glucoside was due to inhibition of iNOS in RAW 264.7 cells without any cytotoxic effects [30]. Moreover, a previous study reported that sericin dose-dependently down-regulated iNOS expression [7]. Therefore, the sericin hydrogel combined with PWCC extract could show a synergistic anti-inflammatory effect.

### 3.7. Effect of Hydrogels and Active Ingredients on Inflammatory Cytokine Gene Expression in LPS-Stimulated RAW 264.7 Cells

The previous results showed that formulation SN6 demonstrated the highest %cell viability and %NO inhibition. Therefore, formulation SN6 was further investigated for its effect on the gene expression of the inflammatory cytokines IL-6, IL-1β, TNF-α, and iNOS in LPS-stimulated RAW 264.7 cell lines using qPCR. Formulation SN6 significantly reduced expression of IL-6, TNF-α, COX-2, and iNOS compared to that of the control (+LPS) and the equivalent hydrogel base without PWCC (SN2), as shown in Figure 4. In the inflammatory process, NO is produced by iNOS using the nitrogen atom from L-arginine [31]. Pro-inflammatory cytokines such as IL-6, IL-1β, and TNF-α are secreted in early-stage inflammation following stimulation with LPS in the RAW 264.7 cell line model [32]. The expression of cytokines IL-6, IL-1β, and TNF-α was induced in RAW 264.7 cells by LPS. Formulation SN6 could reduce the levels of cytokines such as TNF-α, IL-1β, and IL-6 when compared with control. However, formulation SN6 showed no significantly different effect on TNF-α and IL-1β level compared to formulation SN2 and L-NAME. Therefore, formulation SN6 could reduce the excessive inflammatory of all cytokines, which might then be associated with other inflammatory mediators (Figure 4). However, in Figure 4c, PWCCS0.15 showed too high a level of IL-6 for ELISA kit determination. This result might be caused by errors.

A previous study reported that cyanidin-3-glucoside inhibited the expression of iNOS and COX-2 [33]. Moreover, Delphinidin 3-sambubioside, a hibiscus anthocyanin, has been reported to exhibit anti-inflammatory activity by down-regulating the IL-1β and IL-6 cytokines by inhibiting the activities of the NF-kB and AP-1 transcription factors [34]. Anthocyanins from *Trifolium pratense* were also reported to suppress gene expression of TNF-α, IL-1β, and IL-6 cytokines and COX-2 [35]. PWCC crude extract has several other phenolic compounds in addition to anthocyanins [33], which did not affect expression of anti-inflammatory cytokines in vitro, and some anthocyanins, such as pelargonidin, peonidin, and malvidin, which did not inhibit COX-2 [36].

The expression of inflammatory-cytokine genes was also evaluated in LPS-stimulated RAW 264.7 cells (Figure 5). The PWCC extract and hydrogel formulations could reduce the expression of the TNF-α, IL-1β, and IL-6 inflammatory cytokines in LPS-stimulated RAW 264.7 cells compared to the control (+LPS). Formulation SN6 showed a significant reduction in the gene expression of TNF-α, IL-1β, and IL-6 compared to its equivalent hydrogel base (SN2), 0.15% *w*/*v* PWCC extract solution (PWCCS0.15), and L-NAME, as shown in Figure 5. The TNF-α, IL-1β, and IL-6 gene expression results of SN6 correlated with the cytokine level results for SN6-treated LPS-stimulated RAW 264.7 cells in Figure 4. Moreover, the amounts of these cytokines confirmed the decrease in inflammatory cytokines involved in inducing iNOS expression in LPS-stimulated RAW 264.7 cells. Several reports have shown that suppression by anthocyanins occurs via the MAPK and NF-KB pathways [34]. Furthermore, the gene expression results in Figure 5 indicated that SN6 could not down regulate iNOS and COX2 gene expression. Therefore, these effects need to be investigated.

### 3.8. Evaluation of Releasing Anthocyanin Release from Sericin Hydrogel Formulations

The anthocyanin release profiles of hydrogels containing 0.15% *w*/*v* PWCC extract (SN5–SN8) and 0.15% *w*/*v* PWCC extract solution (PWCCS) are shown in Figure 6. All hydrogel formulations released their highest cumulative anthocyanin content within 12 h. Whereas PWCCS provided the highest cumulative anthocyanin level within 2 h. Therefore, hydrogel formulations (SN5, SN6, SN7, and SN8) could prolong the release of anthocyanins. The alginate concentrations in SN5, SN6, SN7, and SN8 were varied at 0, 0.2, 0.3, and 0.4% *w*/*v*, respectively. However, formulations SN5, SN6, SN7, and SN8 showed similar release profiles. The formulation that contained the lowest alginate concentration (SN6) released more anthocyanins than the formulations with higher alginate concentrations (SN7 and SN8) (Figure 6). However, formulation SN5, which contained no alginate, released the highest cumulative amount of anthocyanin.

To evaluate hydrogel drug delivery systems, focusing on the physical and chemical properties of the hydrogel network. The release behavior of hydrogel is affected by the crosslinked networks of polymers, the swelling capacity of the hydrogel, and the types of polymers [3].

Sericin is a hydrophilic polymer chain with a high water content and a high swelling capacity that can perform cross-linking or interaction with other hydrophobic polymers. The swelling of the hydrogel is important to the controlled release of compounds. Initially, anthocyanins rapidly diffused out of the hydrogel before the rate of release slowed. Control of anthocyanin release appears to be influenced by the interaction between sericin and anthocyanin [3].

Several kinetic models were used to predict anthocyanin release from sericin hydrogels (SN5–SN8) (Table 3). The best model was the one with the highest correlation coefficient (R^2^). The release of anthocyanin from all hydrogels (SN5–SN8) followed the Korsmeyer–Peppas model. Whereas the PWCCS showed zero-order kinetics (R^2^ > 0.94) where the release of anthocyanin was independent of its concentration. Furthermore, the release of anthocyanin from hydrogels based on the Korsmeyer–Peppas model (0.45 ≤ *n* ≤ 0.89) describes the drug release as a non-Fickian diffusion mechanism [20].

The release rate of anthocyanin from the polymeric matrix by diffusion can be determined by the porosity of the polymeric matrix. The SEM images of the cross-section scaffold of SN2, SN5, SB6, SN7, and SN8 are presented in Figure 7. The SEM images indicate that the sericin-alginate hydrogels had a highly porous structure. The pore size was calculated using Image J software, and the results showed that SN2 (0.20% alginate without PWCC) showed the largest average pore size at 125.05 ± 4.10 μm. The average pore sizes of SN5, SN6, SN7, and SN8 were 85.20 ± 3.78, 78.20 ± 10.80, 75.00 ± 2.71, and 69.53 ± 5.38 μm, respectively. Therefore, higher concentrations of alginate produced smaller pore sizes in hydrogels. Moreover, the SN6 formulation showed both open and closed pore structures and an increased inter-connection between pores carrying the active anthocyanin. The efficiency release was observed in SEM images.

The results of the swelling study of hydrogels SN5, SN6, SN7, and SN8 are presented in Figure 8. The hydrogels with alginate (SN6, SN7, and SN8) showed a higher percentage of swelling and could preserve the interaction in the hydrogel network better than the hydrogel without alginate (SN5). The interaction in the SN5 hydrogel might be easily broken within 4 h while performing the swelling study. The swelling of the hydrogel is also important for controlled drug release. Initially, swelling allows for rapid diffusion of anthocyanins from the hydrogel since the diffusion release mechanism occurs at the outer surface of the polymer and not in the core porous structure. Therefore, the release occurs rapidly. Later, the compound residing in the core porous structure of the hydrogel is slowly released.

The controlled release of anthocyanin from a hydrogel might be influenced by the interaction between sericin and anthocyanin [3]. Therefore, the mechanism of sustained release from the sericin-alginate hydrogel polymer was via non-erosion of the polymer matrix [20].

### 3.9. FT-IR Spectroscopy

FT-IR spectra of the components of the hydrogels are presented in Figure 9. The FT-IR spectrum of sericin showed functional groups of protein amides at 1655.68, 1554.74, and 1239.72 cm^−1^ referring to amide I, amide II, and amide III, respectively. This indicates α-helix, β-sheet, and a random coil structure, respectively. In addition, the peaks at 3275.72 cm^−1^ (OH groups) and 2935.84 cm^−1^ (C–H bond) were also obtained from the IR spectra of sericin [8,21,27].

Purple waxy corn cob extract (PWCC) showed peaks of anthocyanin at 3343.83, 2923.67, and 1736.75 cm^−1^ referring to OH stretching, CH-stretching, and C=O stretching. The peak at 1463.82 cm^−1^ refers to C−C stretching vibration in the aromatic ring and stretching in the pyran ring is typical of flavonoids. The peaks at 1235.70 and 1147.48 cm^−1^ refer to C−O stretching and C−O−C-stretching, respectively. Minor peaks were observed at 970.42, 920.82, and 820.71 cm^−1^ indicating C−H deformation-aromatic rings [9].

The FT-IR spectrum of alginate showed a broad peak at 3263.07 cm^−1^ referring to OH stretching, peaks at 1593.06 and 1407.07 cm^−1^ referring to COO-asymmetric and symmetric vibration, respectively, and a peak at 1021.68 cm^−1^ referring to C-O stretching. Minor peaks were observed at 947.80, 882.88, and 809.93, indicating C−H bending [5].

The FT-IR spectrum of PVA showed a broad band of 3200–3600 cm^−1^ referring to O–H stretching, a peak at 2910.07 cm^−1^ referring to C–H stretching of CH_2_, and peaks at 1715.07, 1242.06, and 1180.54 cm^−1^ referring to C=O stretching, C−C stretching, and C−OH stretching, respectively. Minor peaks were observed at 947.311 cm^−1^ referring to CH_2_ rocking and 841.90 cm^−1^ referring to C−C stretching [37,38].

The FT-IR spectra of hydrogels (SN2 and SN6) showed peak shifting that indicated the interactions between the components through functional groups. Both SN2 and SN6 showed a high-intensity broad band in the region of 3200–3600 cm^−1^ (O–H stretching), which was from the overlapping of sericin, alginate, and PVA.

The important peaks of amide I, amide II, and amide III at 1655.68, 1554.74, and 1239.72 cm^−1^ in sericin were shifted to 1625.59, 1522.89, and 1258.63 cm^−1^ in SN2 and 1629.68, 1522.36, and 1261.66 cm^−1^ in SN6. These shifts confirmed that the interaction occurred at the amide group (−(C=O)−NH−) and amino groups of sericin. Furthermore, the absence or presence of peaks indicates an interaction or change in structure. The absence of the 1655.68 cm^−1^ sericin peak indicates the absence of α-helix in hydrogels. Whereas the presence of peaks at 1625.59 cm^−1^ for SN2 and 1629.68 cm^−1^ for SN6 refers to β-sheet structures, which might be involved with interaction in hydrogels [39,40].

The interaction between sericin and alginate in the hydrogels showed that the carboxylic peak (at 1593.06 cm^−1^) of alginate and the amide II group (at 1554.74 cm^−1^) of sericin were absent. The interaction between sericin and alginate in the hydrogel showed the peak of the N-H bending vibration band of sericin was shifted from 1554.74 cm^−1^ to 1522.89 cm^−1^ and 1522.36 cm^−1^ for SN2 and SN6, respectively. Therefore, sericin’s inherently ordered structure was changed in hydrogel formulations due to the ionic and intramolecular bonds formed between the positively charged amide II group of sericin and the negatively charged carboxyl group of alginate (Figure 10b) [41].

The amide I and III peaks at 1625.59 and 1258.63 cm^−1^ in SN2 were shifted to 1629.68 and 1261.66 cm^−1^ in SN6 containing PWCC. Whereas the peaks of amide II in SN2 (1522.89 cm^−1^) and SN6 (1522.36 cm^−1^) were only slightly different. The results indicated that the interaction of anthocyanin and sericin occurred at amide I and III, in which the phenolic OH of anthocyanin interacted with the C=O (amide) of sericin via hydrogen bonds (Figure 10a).

For PVA interaction, the sharp peak at 2910.07 cm^−1^ (C–H stretching of the CH_2_ group) of PVA was shifted to a lower intensity peak at 2941.75 cm^−1^ for SN2 and a peak at 2943.78 cm^−1^ for SN6 in hydrogel spectra. Whereas the peak at 1715.07 cm^−1^ (C=O stretching) of PVA became less intense and overlapped amide I in hydrogels (SN2 and SN6), which indicated no interaction between the C=O group of PVA and the amide I group of sericin. [39]. In addition, the strong intensity peak at 1180.54 cm^−1^ (C−OH stretching of secondary alcohol) of PVA was shifted to a low intensity peak at 1080.54 cm^−1^ in SN2 and 1086.60 cm^−1^ in SN6, which might indicate the interaction of the −OH of PVA with the C=O of alginate via H-bond (Figure 11) [42].

A previous study reported that interactions between alginate and anthocyanin molecules occurred via charge-charge interaction between the negatively charged carboxylic groups of the polymeric backbone of alginate and the flavylium cation of anthocyanin [43]. However, it may be difficult to observe the alginate peaks in SN6 that indicate these structures clearly since the alginate in the formulation was low in concentration compared to sericin. Therefore, amides I, II, and III are mostly applicable to indicate interactions in hydrogels.

As a result of the releasing profile, anthocyanin was gradually released from hydrogels, which upon the swelling of the hydrogel, changed the structure of sericin from β-sheet to random coils. Therefore, the sericin hydrogel could prolong anthocyanin release through anthocyanin-sericin interactions for diffusion through the crosslink-network hydrogel.

### 3.10. Stability Evaluation

The pH of all hydrogel formulations (SN2, SN5, and SN6) showed no significant differences before and after storage under heating-cooling conditions. Whereas only SN6 contained 0.20% alginate and 0.15% PWCC showed no significant difference in viscosity. The amount of anthocyanin recovered from sericin hydrogels (SN5 and SN6) showed no significant differences. In contrast, the anthocyanin content of PWCC extract in solution decreased significantly after storage, as shown in Table 4. Therefore, the SN6 formulation could provide the high stability of anthocyanins and improve other physical properties after storage conditions. 

## 4. Conclusions

Sericin-alginate hydrogel formulations containing purple waxy corn (*Zea mays* L.) extract as the active ingredient indicated sustained release of anthocyanins following Korsmeyer–Peppas kinetics. FT-IR confirmed the control release of anthocyanin occurred due to interactions between sericin and anthocyanin by hydrogen bonding. The optimised hydrogel formulation, SN6 (2.00% *w*/*v* sericin, 0.20% *w*/*v* alginate, and 0.15% *w*/*v* purple waxy corn extract), showed in vitro anti-inflammatory activity in LPS-stimulated RAW 264.7 cells through the inhibition of nitric oxide production and down-regulation of IL-6, IL-1β, and TNF-α expression without cytotoxicity. The hydrogel formulation SN6 was highly stable under heating-cooling storage conditions. Thus, the developed hydrogel formulation provided a sustained release of anthocyanin for anti-inflammatory activities. These results will provide the basis for in vivo and clinical trials in the future.

## Figures and Tables

**Figure 1 pharmaceutics-14-00577-f001:**
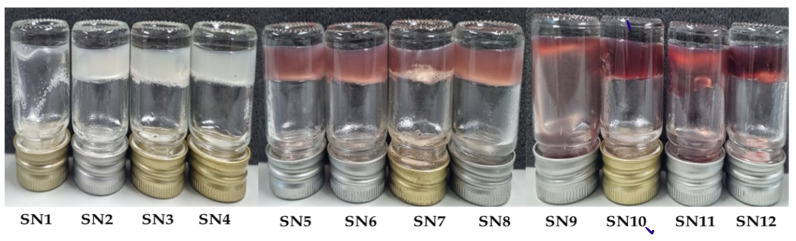
Characteristics of sericin-hydrogel formulations (SN1–SN12).

**Figure 2 pharmaceutics-14-00577-f002:**
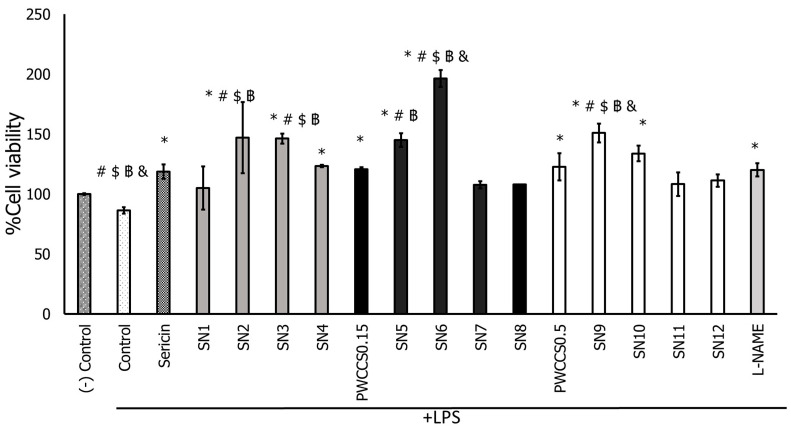
Effect of sericin, purple waxy corn cob extract solution 0.15 and 0.50% (PWCCS0.15 and PWCCS0.5), sericin-hydrogel formulations (SN1–SN12), and L-NAME 200 μM (L-NAME) on cell viability of RAW 264.7 cells treated with LPS. Data represent the mean ± SEM of three replicates. Statistical significance was evaluated by Dunnett post-hoc test (* *p* < 0.05 compared with LPS-treated cells (control), ^#^ *p* < 0.05 compared with L-NAME 200 μM, ^$^ *p* < 0.05 compared with PWCCS0.15, ^฿^ *p* < 0.05 compared with sericin, ^&^ *p* < 0.05 compared with PWCCS0.5).

**Figure 3 pharmaceutics-14-00577-f003:**
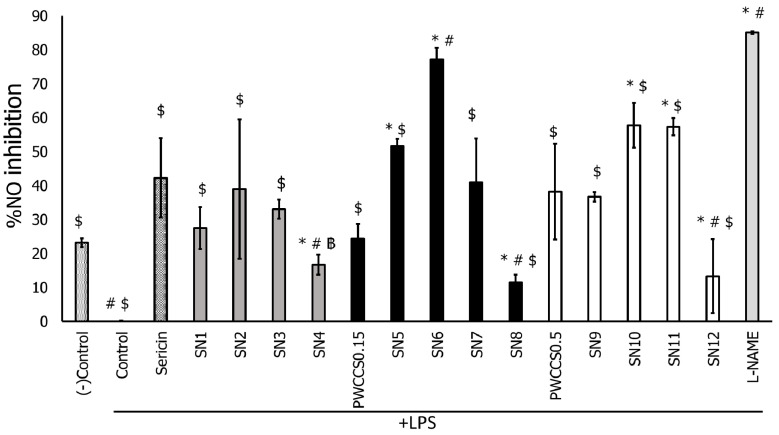
Effects of sericin, sericin-hydrogel formulations (SN1–SN12), L-NAME 200 μM, purple waxy corn cob extract solution 0.15% (PWCCS0.15), and 0.50% (PWCCS0.5) on nitric oxide production by LPS-stimulated RAW 264.7 cells. Data represent the mean ± SEM of three replicates. Statistical significance was evaluated by Dunnett post-hoc test, * *p* < 0.05 compared with (−)control, ^#^ *p* < 0.05 compared with SN2, ^฿^ *p* < 0.05 compared with sericin, ^$^ *p* < 0.05 compared with L-NAME 200 μM.

**Figure 4 pharmaceutics-14-00577-f004:**
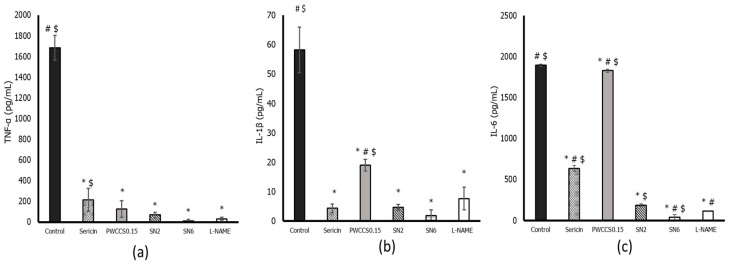
Inflammatory cytokine levels of TNF-α (**a**), IL-1β (**b**), and IL-6 (**c**) in LPS-stimulated RAW 264.7 cells. Data represent the mean ± SEM of three replicates. Statistical significance was evaluated by Dunnett post-hoc test, * *p* < 0.05 compared with control (LPS-treated group), ^#^ *p* < 0.05 compared with SN2 and ^$^ *p* < 0.05 compared with L-NAME.

**Figure 5 pharmaceutics-14-00577-f005:**
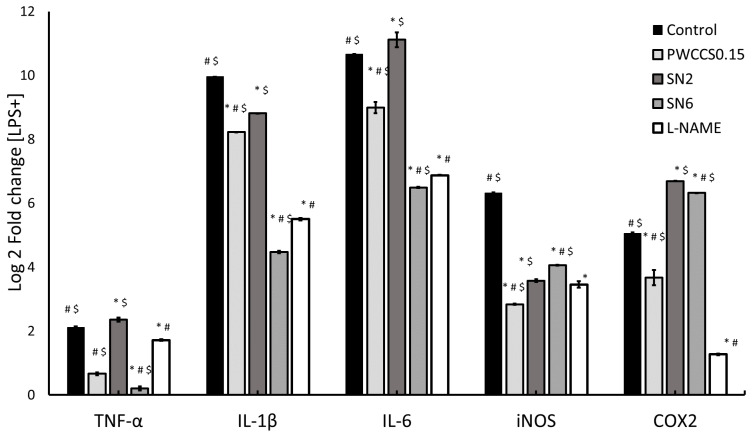
Inflammatory gene expression in LPS-stimulated RAW 264.7 cells. Data are relative expression levels of TNF-α, IL-1β, IL-6, iNOS, and COX-2 from control (+LPS), PWCSS0.15, SN2, SN6, and L-NAME. Data represent the mean ± SEM of three replicates. Statistical significance was evaluated by Dunnett post-hoc test, * *p* < 0.05 compared with control (+LPS), ^#^ *p* < 0.05 compared with SN2 and ^$^ *p* < 0.05 compared with 200 μM L-NAME.

**Figure 6 pharmaceutics-14-00577-f006:**
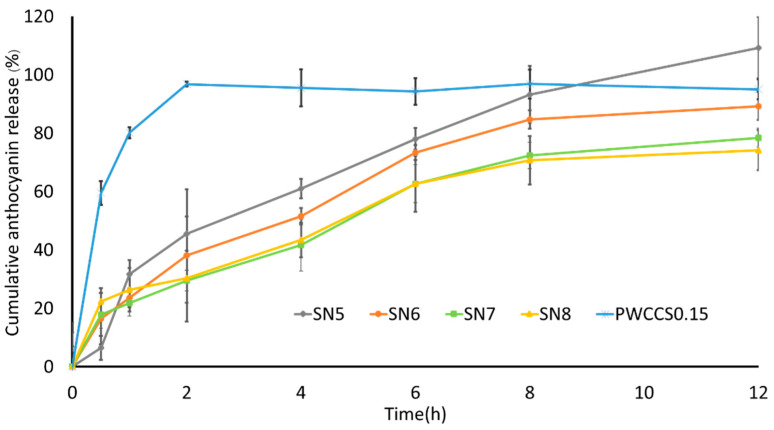
The release profiles of 0.15% *w*/*v* PWCC extract solution (PWCCS0.15) compared with sericin hydrogels (SN5–SN8) contained 0.15% PWCCS.

**Figure 7 pharmaceutics-14-00577-f007:**
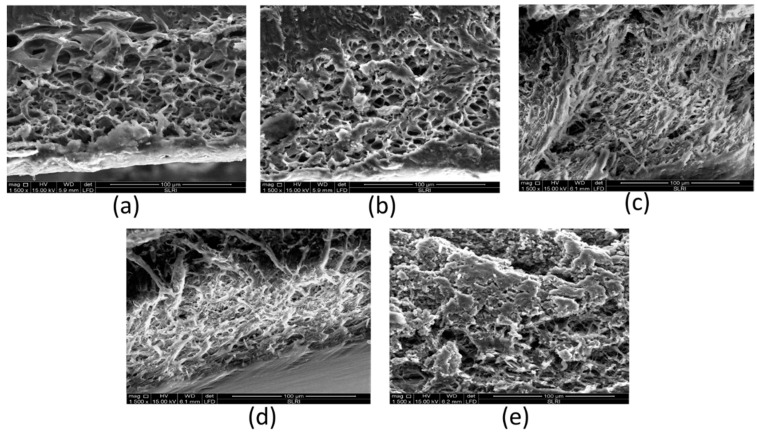
SEM images of cross-section scaffold; SN2 (**a**), SN5 (**b**), SN6 (**c**), SN7 (**d**), and SN8 (**e**).

**Figure 8 pharmaceutics-14-00577-f008:**
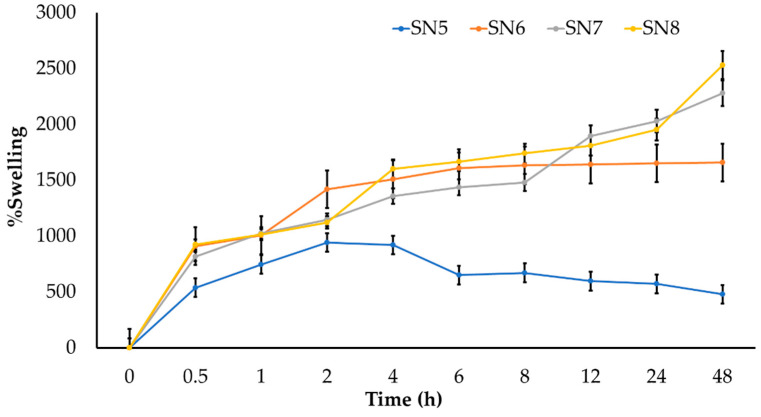
Swelling studies of hydrogels (SN5–SN8) in PBS pH 7.4.

**Figure 9 pharmaceutics-14-00577-f009:**
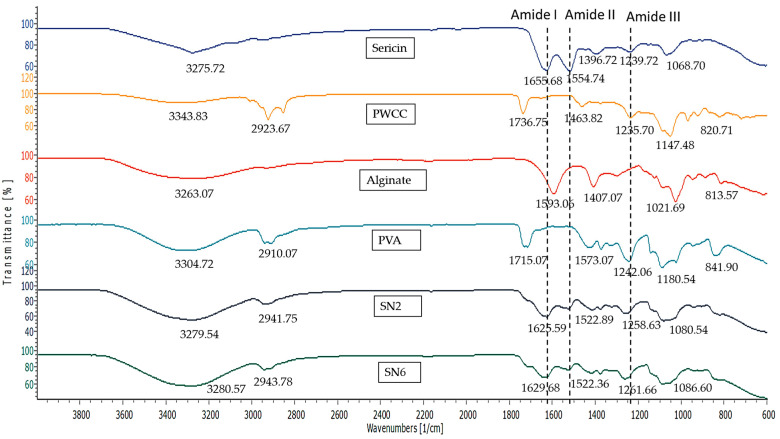
Infrared spectra of Sericin, Alginate, PWCSS, and formulations SN2 and SN6.

**Figure 10 pharmaceutics-14-00577-f010:**
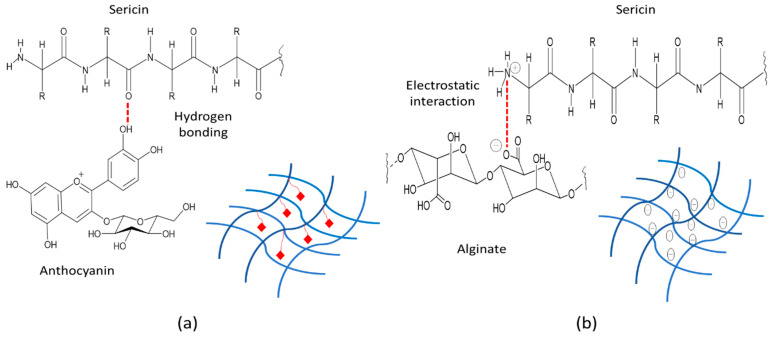
Illustration of sericin, alginate, and anthocyanin interactions (**a**) Sericin-anthocyanin interaction: hydrogen bonding (**b**) Sericin–alginate interaction: electrostatic interaction.

**Figure 11 pharmaceutics-14-00577-f011:**
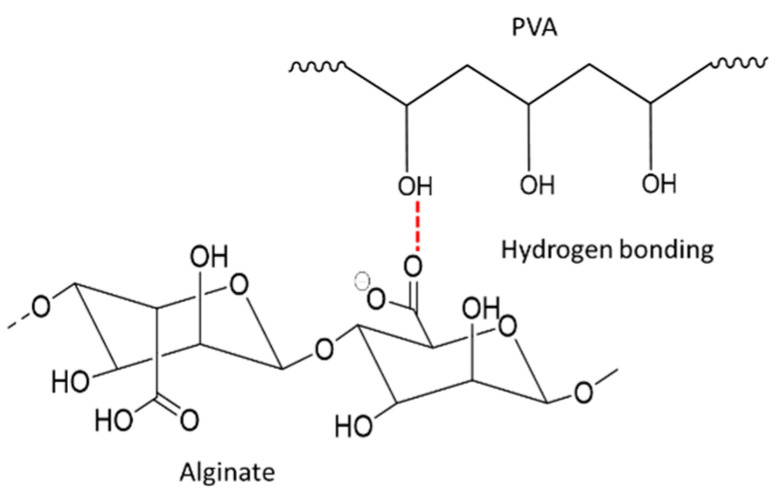
Proposed illustration of alginate interaction PVA via H-bond.

**Table 1 pharmaceutics-14-00577-t001:** The composition of the hydrogel formulations.

Formulation	5% *w*/*v* Sericin (mL)	10% *w*/*v* PVA (mL)	5% *w*/*v* Alginate (mL)			10% *w*/*v* PWCCS (mL)			Final Volume (mL)	Final Conc. of Alginate (%)	Final Conc. of PWCC (%)
SN1	2.00	2.00	-	-	-	-	-	-	5.00	0.00	0.00
SN2	2.00	2.00	0.20	-	-	-	-	-	5.00	0.20	0.00
SN3	2.00	2.00	-	0.30	-	-	-	-	5.00	0.30	0.00
SN4	2.00	2.00	-	-	0.40	-	-	-	5.00	0.40	0.00
SN5	2.00	2.00	-	-	-	-	0.075	-	5.00	0.00	0.15
SN6	2.00	2.00	0.20	-	-	-	0.075	-	5.00	0.20	0.15
SN7	2.00	2.00	-	0.30	-	-	0.075	-	5.00	0.30	0.15
SN8	2.00	2.00	-	-	0.40	-	0.075	-	5.00	0.40	0.15
SN9	2.00	2.00	-	-	-	-	-	0.25	5.00	0.00	0.50
SN10	2.00	2.00	0.20	-	-	-	-	0.25	5.00	0.20	0.50
SN11	2.00	2.00	-	0.30	-	-	-	0.25	5.00	0.30	0.50
SN12	2.00	2.00	-	-	0.40	-	-	0.25	5.00	0.40	0.50

**Table 2 pharmaceutics-14-00577-t002:** Physical characteristics and total anthocyanin content of sericin hydrogels and extracts.

Code	Conc. of PWCC (%)	pH	Viscosity (Pa·s)	Total Anthocyanin (mg C3GE/L)	Appearance
PWCCS	0.15	-	-	43.75 ± 1.44 ^a^	
PWCCS	0.50	-	-	101.93 ± 19.54 ^b^	
SN1	-	6.55 ± 0.00	35.52 ± 0.00 ^e^	-	White and translucent
SN2	-	6.62 ± 0.10	37.51 ± 0.00 ^g^	-	White and translucent
SN3	-	6.51 ± 0.01	38.52 ± 0.00 ^i^	-	White and translucent
SN4	-	6.48 ± 0.09	39.55 ± 0.00 ^j^	-	White and translucent
SN5	0.15	6.57 ± 0.00	35.39 ± 0.00 ^c^	41.48 ± 6.62 ^a^	Light purple and homogenous
SN6	0.15	6.71 ± 0.28	35.50 ± 0.00 ^d^	39.21 ± 3.26 ^a^	Light purple and homogenous
SN7	0.15	6.58 ± 0.00	36.52 ± 0.00 ^f^	41.81 ± 4.06 ^a^	Light purple and homogenous
SN8	0.15	6.65 ± 0.33	37.55 ± 0.00 ^h^	40.08 ± 8.01 ^a^	Light purple and homogenous
SN9	0.50	6.47 ± 0.00	34.27 ± 0.00 ^a^	107.61 ± 11.98 ^b^	Dark purple
SN10	0.50	6.45 ± 0.05	34.49 ± 0.00 ^b^	102.87 ± 1.53 ^b^	Dark purple
SN11	0.50	6.49 ± 0.05	35.50 ± 0.00 ^d^	100.19 ± 19.22 ^b^	Dark purple
SN12	0.50	6.61 ± 0.18	36.52 ± 0.01 ^f^	100.13 ± 2.52 ^b^	Dark purple

These data represent the mean ± SEM of three replicates. ^a–j^ letters indicate significant differences in the same column at *p* < 0.05 (by one-way ANOVA).

**Table 3 pharmaceutics-14-00577-t003:** Kinetic model prediction of anthocyanin releasing from 7sericin hydrogels (SN5–SN8) and 0.15% PWCC (PWCCS0.15).

Sample	Conditions	Zero-Order		First-Order		Higuchi		Korsmeyer–Peppas		
		R^2^	K_0_	R^2^	K_1_	R^2^	KH	R^2^	n	KP
PWCCS0.15	pH5.5, 37 °C	0.94	23.71	0.90	0.13	0.74	26.64	0.84	0.23	75.20
SN5		0.97	7.05	0.89	0.05	1.00	32.09	1.00	0.50	31.67
SN6		0.89	6.13	0.81	0.05	0.96	28.53	0.98	0.56	24.72
SN7		0.96	5.16	0.94	0.06	0.96	25.22	0.98	0.56	21.11
SN8		0.91	3.34	0.88	0.05	0.96	15.37	0.96	0.49	15.78

PWCCS0.15 is 0.15% *w*/*v* PWCC extract solution and SN5–SN8 are sericin hydrogel formulations contained 0.15% PWCCS. K_0_ is a constant of zero-order, K_1_ is a constant of first-order, KH is a constant of Higuchi model, KP is a constant of Korsmeyer–Peppas model and n is the diffusional exponent characteristic of the release from Korsmeyer–Peppas model.

**Table 4 pharmaceutics-14-00577-t004:** The stability tests for determination of pH properties, viscosity (Pa·s), and anthocyanin content remaining comparison before and after heating-cooling storage condition.

Sample	Conc. of PWCCS (%)	pH		Viscosity (Pa·s)		Total Anthocyanin (mg C3G/L)	
		Before	After	Before	After	Before	After
PWCCS	0.15	-	-	-	-	43.75 ± 1.44 *	13.15 ± 0.28 *
SN2	0.00	6.62 ± 0.10	6.54 ± 0.00	37.51 ± 0.00 *	36.55 ± 0.01 *	-	-
SN5	0.15	6.57 ± 0.00	6.57 ± 0.06	35.39 ± 0.00 *	37.52 ± 0.01 *	41.48 ± 6.62	36.03 ± 0.06
SN6	0.15	6.71 ± 0.28	6.49 ± 0.05	35.57 ± 0.00	35.57 ± 0.00	39.20 ± 3.25	37.26 ± 0.00

These data represented the mean ± SEM of three replicates. * *p* < 0.05, comparison between before and after storage condition, by paired sample *t*-test.

## Data Availability

Not applicable.

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
