# Peer review of "Development of a Sericin Hydrogel to Deliver Anthocyanins from Purple Waxy Corn Cob (Zea mays L.) Extract and In Vitro Evaluation of Anti-Inflammatory Effects"

_pharmaceutics, 2022, doi:10.3390/pharmaceutics14030577_

Round 1

Author Response

Please see the attached file and the link for supporting materials.

https://www.dropbox.com/sh/lab40krnsc9a9yp/AAAfsJo3vqOhz8UrO7v1gaOGa?dl=0

Reviewer 2 Report

Comments to the authors:

The manuscript is entitled “Development of a sericin hydrogel to deliver anthocyanins  from purple waxy corn cob (Zea mays L.) extract and in vitro evaluation of anti-inflammatory effects". Nattawadee Kanpipit et al. developed and evaluated  Sericin-alginate hydrogel formulations with purple waxy corn (Zea mays L.) cob extract PWCC) for topical anti-inflammatory application. Physical properties such as viscosity, pH, and anthocyanin release were examined and in vitro anti-inflammatory  activities such as NO inhibition and IL-6, IL-1β, TNF-α, iNOS, and COX-2 expression were evaluated in LPS-stimulated RAW 264.7 murine macrophages.

This manuscript can be accepted for publication in Pharmaceutics Journal, however, it needs some adjustments. I recommend Accept after major revision. After a critical evaluation of the manuscript, some comments are done as follows:

Abastract:

Line 23: “by chemical crosslinking through ionic interaction of the polymers combined with anthocyanin”. Isn't ionic interactions physical crosslinking? Please check it. Another question, I could not understand why using PVA for the hydrogel´s formation. For the same is not mentioned anywhere. Only in the supplementary material, which was not even commented on at work.

Keywords:

I suggest that you remove from the keywords the words that are in the title paper´s title.

Introduction:

I think the link between the paragraphs is necessary. Please check.

Line 64: “In this study, we have developed a hydrogel formulation to deliver anthocyanins from natural sources. Purple waxy corn (Zea mays L.) cob extract, a natural enriched source of anthocyanins, was utilized as the model for the formulation”. In my opinion, this part is placed at the end of the introduction, being the objective of the work.

Line 67: “KND” what does it mean? It does not appear previously anywhere.

I think we need to talk about PVA, its properties and why its use. In table 1 it appears as part of the hydrogels’ formation, however it is not mentioned anywhere. With that, my doubt, was the PVA used at work, and if so, why?

Line 84: “The interaction of sericin and alginate generates a network and forms a porous structure suitable for drug delivery systems with improved physical properties”. What kind of interaction? Does PVA not participate in the formation of hydrogels? What is the use of PVA? If the interaction is physical, why do you talk about chemical crosslinking in the abstract? For me, this is part of the results and discussion, not the introduction.

Materials and methods:

Line 105: “The silkworm cocoons were cut into small square pieces” What size was cuted?

Results and Discussion

Table 1: Standardize the font. Why were these precursor concentrations chosen? Was it based on the literature or did you determine it yourself?

Line 222: “A previous study found the highest anthocyanin content in the corn cob part of purple waxy corn KND strain by 50%  ethanol extraction”. What value?

Line 275: “Therefore, using 2%w/v sericin in hydrogels”. In table 1, it says that 5% w/v of sericin was used. Please check it.

Line 396: “The correlation coefficients (R2) indicated that sericin hydrogel SN6 396 was followed the Higuchi model more than zero- and first-order models”. This is not what Table 3 shows. In which the zero-order model had a better fit than Higuchi. Is there a difference between these r2 values? The values ​​are very close. Why aren’t the other hydrogels that were released in the table?

If you look critically at the release data in Fig. 6 there are 2 release steps. Being the first stage in 1 hour approximately and the second from that. Wouldn't it be necessary to check each step and see which model applies to each one? Please check it

Line 400: “The release rate of anthocyanin from the polymeric matrix by diffusion can be determined by the porosity of the polymeric matrix”. What is the % porosity, or how are hydrogels known to be porous? I think it is necessary to do an analysis of Scaninig electronic microscopy or some analysis that shows the % of porosity of the material.

Line 401: “This diffusion release mechanism occurs 401 at the outer surface of the polymer, not in the core porous structure”. I do not agree, because the model that best fits is zero order, as shown in Table 3.

Figure 7: Please improve the resolution of the figure.

FTIR: Is there any ionic or electrostatic interaction? If yes, what are the pKAs of the precursors? And which bands demonstrate this.

Placing the wavenumber instead of the groupings makes it easier to see and identify it on the work, as the groupings are written in the text.

Confusing the FTIR discussion. I suggest you talk about the precursors first and then show the interactions that show the formation of the hydrogels as well as the presence of PWCCS, sericin, PVA and alginate.

I do not understand why the PVA is in the supplementary material if the presence of the PVA was not mentioned or discussed throughout the work. I finish reading the work and I can't tell if the PVA was actually used in the work.

Improve the resolution of figures in the supplementary material.

Author Response

Please see the attached file and the share link. 

https://www.dropbox.com/sh/lab40krnsc9a9yp/AAAfsJo3vqOhz8UrO7v1gaOGa?dl=0

Reviewer 3 Report

In this manuscript ‘Development of a sericin hydrogel to deliver anthocyanins 2 from purple waxy corn cob (Zea mays L.) extract and in vitro evaluation of anti-inflammatory effects’, the authors describe the synthesis and characterization of a novel sericin alginate hydrogel and performance as delivery vehicle of anthocyanins.

Overall the experimental approach is reasonable and also the manuscript is well-drafted. From the application perspective, the results and discussions are nicely presented.

I have few suggestions and corrections, which the authors should address:

  1. Line 67-68: Source of purple waxy corn strain should be part of methods section. Remove from Introduction.
  2. Line 105: In 1-2 sentences, provide details of the high pressure and temperature extraction method.
  3. Lie 117: Briefly describe the principle of the pH-differential method.
  4. In the FTIR spectra of Alginate and PWCCS (Figures 7, S4 and S5), minor peaks in the range of 950 to 800 cm-1 are not indexed. These indicates a minor fraction of calcium ions are complexed with the alginate and PWCCS molecules, and /or carbonate ions. In practice, it is also challenging to completely isolate macromolecules such as alginate from metal divalent ions in nature. For indexing these FTIR bands, see: pH-dependent schemes of calcium carbonate formation in the presence of alginates. Crystal Growth & Design, 16(3), 1349-1359.
  5. Figure 8: Could there be a strong/electrostatic interaction also between anthocyanin and alginate molecules? This should be briefly discussed. See: "Formation and characteristics of alginate and anthocyanin complexes." International Journal of Biological Macromolecules 164 (2020): 726-734.

Author Response

(The authors gave the same response as above.)

Reviewer 4 Report

The article entitles Development of a sericin hydrogel to deliver anthocyanins from purple waxy corn cob (Zea mays L.) extract and in vitro evaluation of anti-inflammatory effects, shows the results of a hydrogel composed by different polymers to deliver topically an anti-inflammatory compound of natural origin. Although the article is interesting, the authors should increase hydrogel characterization and should improve results and discussion of some non-expected results. I recommend resubmitting the manuscript after a deep revision and I recommend completing the article with some more experimental.

Some specific comments:

Section 2.1. LPS is repeated.

Section 2.2.1 please increase the description of the extraction procedure. If it was previously described, please include the reference.

Section 2.5. please include the amount of formulation placed in the donor compartment of Franz cells.

Line 147. Korsmeyer-Peppas (KP) equation is not the square root of time (this is only for Higuchi’s model). KP is the “n” potency.

Line 157. L-NAME was used as positive control of cytotoxicity? Of proliferation? The results of cell viability with L-NAME are higher compared with control

Section 2.11. the accelerated condition is usually 40ºC/75% HR (not 45ºC) and it is usually carried out for 6 months. The stability study at 5ºC is not an accelerated condition. In addition, if the intended long-term conditions are 5ºC then the accelerated stability condition is 25ºC/25% HR, according to the ICH guideline.

Line 207. Turkey is usually carried out as post-hoc test, as well as Dunnett test. Which one did you apply? ANOVA is ANOVA not a Turkey test.

Line 215. 14.5% w/? there are unit left.

Line 217. It sounds a little bit unrealistic the molecular weight result of 37-250 kDa. This result should be further discussed or explained. What is the theoretical molecular weight? The purification process maybe it not very selective. If it is not selective the reproducibility of the formulations could be low.

The visualization of table 2 is poor, maybe caused by the conversion to pdf.

The characterization of the hydrogels should be completed with swelling and porosity studied. SEM microscopy is also useful to corroborate the porosity of the hydrogels.

Section 3.6.1 and figure 2. In the figure why the control is not 100%? Usually, the results are referend to control value. According to section 2.7.1, the cytotoxicity study is not performed with LPS stimulated cells, but in section 3.6.1 does. In addition, results without LPS are not reported in the text. Finally, the alginate and/or extract PWCC do not have a dose-concentration effect on the cell proliferation, the effect on proliferation is difficult be assigned to these components due to the lack of response with the concentration increase.

Figure 3. the results and discussion should be increased including the lack of inhibition of formulation SN8 and SN12 (with the same concentration of extract than other formulations but without effect). Again, there is a lack on dependency between the extract concentration and its effect. Also, it was not discussed the effect of different levels of alginate, that seems that block the inhibition of NO synthesis.

Figure 5. The pure extract seems that increase the inflammatory cytokines IL1b, IL6 and COX2. What could be the reason? Is the extract proinflammatory? Also, formulation SN6 increase the COX2 levels.  

Figure 6. the curve of the pure extract is not very common. If the plot is cumulative the release level of the extract should be constantly at 100% after arriving to the plateau. The reduced levels after the 2h could be due to solubility or stability issues in the receptor medium for example.

Table 3. R2 value is higher for zero order compared to Higuchi (in the formulation SN6), so the discussion is not adequate. What “n” stands for? There is not comments about the kinetic model of the solution.

Author Response

(The authors gave the same response as above.)

Round 2

Reviewer 2 Report

The authors adapted the text as requested by the reviewers and also performed analyzes that were requested. Therefore, this work is ready to be published in its present form.

Author Response

Dear Reviewer,

I would like to thank for your valuable comments that help to improve this manuscript. 

Best regards,

Suthasiee Thapphasaraphong

Author Response

Dear Reviewer,

We have already revised this manuscript according to your suggestion. Please see the attached file.

We would like to thank for your valuable comments and suggestion that could help to improve this manuscript.

Best regards,

Suthasinee Thapphasaraphong
